# Exogenous Regulators Enhance the Yield and Stress Resistance of Chlamydospores of the Biocontrol Agent *Trichoderma harzianum* T4

**DOI:** 10.3390/jof8101017

**Published:** 2022-09-27

**Authors:** Xiaochong Zhu, Yaping Wang, Xiaobing Wang, Wei Wang

**Affiliations:** State Key Laboratory of Bioreactor Engineering, East China University of Science and Technology, Shanghai 200237, China

**Keywords:** *Trichoderma harzianum* T4, chlamydospore, exogenous regulators, stress resistance, lipid metabolism, unsaturated fatty acids

## Abstract

*Trichoderma* strains have been successfully used in plant disease control. However, the poor stress resistance of mycelia and conidia makes processing and storage difficult. Furthermore, they cannot produce chlamydospores in large quantities during fermentation, which limits the industrialization process of chlamydospore preparation. It is important to explore an efficient liquid fermentation strategy for ensuring chlamydospore production in *Trichoderma harzianum*. We found that the addition of mannitol, glycine betaine, and N-acetylglucosamine (N-A-G) during liquid fermentation effectively increases the yield of chlamydospores. Furthermore, we provided evidence that chlamydospores have stronger tolerance to high temperature, ultraviolet, and hypertonic stress after the addition of mannitol and trehalose. Lipids are an important component of microbial cells and impact the stress resistance of microorganisms. We studied the internal relationship between lipid metabolism and the stress resistance of chlamydospores by detecting changes in the lipid content and gene expression. Our results showed that mannitol and trehalose cause lipid accumulation in chlamydospores and increase the unsaturated fatty acid content. In conclusion, we verified that these exogenous regulators increase the production of chlamydospores and enhance their stress resistance by regulating lipid metabolism. In addition, we believe that lipid metabolism is an important part of the chlamydospore production process and impacts the stress resistance of chlamydospores. Our findings provide clues for studying the differentiation pathway of chlamydospores in filamentous fungi and a basis for the industrial production of chlamydospores.

## 1. Introduction

*Trichoderma* strains are among the most studied biocontrol fungi and have been successfully used in field disease control and as crop fertilizers [1]. They grow rapidly, induce plant resistance, control pathogens by producing a variety of secondary metabolites [2], and antagonize plant pathogens through spatial and nutritional competition [3]. *Trichoderma*
*harzianum* T4 is an efficient biocontrol strain isolated from soil in our laboratory; it can effectively control the diseases caused by *Botrytis cinerea* and *Fusarium oxysporum* [4]. In addition to effectively controlling field diseases, *T. harzianum* T4 can significantly affect rhizosphere soil bacterial communities in a short time and promote the production of plant probiotics [5,6], thereby increasing crop yield.

*Trichoderma* has good biocontrol potential, but the poor stress resistance of mycelia and conidia makes their production and storage difficult [4]. Under adverse conditions, *Trichoderma* forms chlamydospores, characterized by slow metabolism, thickened cell walls, and the accumulation of compatible substances, such as glycerol and mannitol [7]. Research has shown that *T. harzianum* conidia can survive in the soil for 130 days [8], whereas chlamydospores can survive for 16 months [9]. Therefore, a preparation that comprises chlamydospores may guarantee a prolonged shelf life.

Mannitol, glycine betaine, N-acetylglucosamine (N-A-G), and trehalose can be used as stress-resistant substances to protect living organisms from environmental pressures [10], such as high temperature [11,12], low temperature [13], hypertonic stress [14,15], and oxidation stress [16]. Research has reported their involvement in the formation of fungal spores. Mannitol and trehalose are important components of the intracellular contents of spores [17,18]. While N-A-G represents the fundamental product for chitin biosynthesis, *Trichoderma* uses N-A-G for chitin biosynthesis. Glycine betaine maintains the fluidity of the cell membrane and promotes the synthesis of specific fatty acids under low temperatures [19]. The production of chlamydospores is accompanied by intracellular energy storage and fat accumulation; hence, lipid metabolism is fundamental in chlamydospore production.

In this study, we determined whether the exogenous regulators changed the chlamydospore production and stress tolerance of *T. harzi**anum* T4 during liquid fermentation. The effects of several regulators on lipid accumulation in chlamydospores were investigated by confocal laser scanning microscopy (CLSM) and transmission electron microscopy. The differences in lipid metabolism were investigated by measuring the intracellular lipid content and the expression of key genes. The results provide clues for studying the differentiation pathway of chlamydospores in filamentous fungi and a basis for obtaining industrial preparations with increased chlamydospore content.

## 2. Materials and Methods

### 2.1. Fungal Strains and Media

The wild-type *T. harzianum* T4 strain was used in this study and was stored in our laboratory. A conidial suspension of *T. harzianum* T4 was stored in 20% glycerol at −80 °C. For the acquisition of conidia, *T. harzianum* T4 was inoculated and cultured in potato dextrose agar (PDA) for 3 days, and mycelia were scraped and suspended in sterile water, fully shaken, and filtered with three layers of gauze for backup. The initial culture medium of chlamydospore preparation (IMC) consisted of soluble starch (2.39%), calcium carbonate (0.1%), wheat bran (0.48%), ammonium sulfate (0.2%), potassium dihydrogen phosphate (0.1%), and Tween 80 (0.5%).

Mannitol, glycine betaine, and N-A-G were purchased from Shanghai Macklin Biochemical Co., Ltd. Unless otherwise stated, other chemicals were purchased from Shanghai Titan Scientific Co., Ltd. (Shanghai, China).

### 2.2. Effects of Exogenous Regulators on the Production of Chlamydospores in T. harzianum T4

As stress-resistant regulators, mannitol, glycine betaine, N-A-G, and trehalose were added to IMC at three concentrations of 1%, 2%, and 3% to screen the optimal addition amount. IMC without additives was used as the control group. The conidial suspension was adjusted to 1 × 10^6^ cfu/mL with sterile water and inoculated in IMC (1% inoculum), cultured at 30 °C, and shaken at 200 rpm for 168 h. A total of 1 mL of fermentation solution was poured into the centrifuge tube and ground with a handheld cell tissue grinder for 60 s and then with an ultrasonic crusher for 5 min to evenly distribute the chlamydospores. Chlamydospore production was determined using a blood-cell-counting plate.

### 2.3. Survival Tests

We tested the stress resistance of conidia, primitive chlamydospores, and regulated chlamydospores. The preparation of the conidial suspension is described earlier. The preparation method of the chlamydospore suspension was as follows: The fermentation liquid was vortexed for 5 min, followed by ultrasonic treatment for 10 min; the mycelium was removed by filtration using double-layer lens-wiping paper. The chlamydospore suspension was cleaned with PBS three times to remove the residual medium components. All spore suspensions were diluted to 10^2^ cfu/mL and treated with UV, heat, and hypertonic stresses. The spore suspensions were irradiated under UV light for 20, 40, 60, and 80 s. For heat treatment, the spore suspensions were placed in a water bath under high-temperature stress (55 °C) for 1, 3, 5, and 7 min [20]; each group was pipetted 100 μL and coated on PDA medium. The germination rate was determined after 24 h. For hypertonic stress, 100 μL of the diluted spore suspension was sucked and coated on PDA medium at concentrations of 0%, 3%, 6%, and 9% NaCl. The germination rate was determined after 24 h [17]. We set up a control group without stress in each group of the stress experiments. Each experiment was repeated at least three times.

### 2.4. Effects of Exogenous Regulators on Lipid Accumulation in the Chlamydospores of T. harzianum T4

The total fat content was determined according to the modified method of Wang et al. [21]. In brief, the chlamydospores in each group were quantified to 1 × 10^8^ cfu/mL; 1 mL of chlamydospore suspension was crushed with an ultrasonic crusher for 120 s (power 85%, ultrasonic 5 s, stop for 5 s); and 3 mL of a methanol chloroform extract (methanol:chloroform = 2:1) was added, mixed, and shaken well for 5 min and then centrifuged at 4500× *g* and 4 °C for 10 min. The organic phase was blow-dried with nitrogen to obtain the total fat content.

The triglyceride (TG) content was determined according to Hu et al.’s method [22] with slight modifications. Briefly, 400–500 × 10^4^ chlamydospore cells were sampled, the supernatant was discarded after centrifugation, 1 mL of extraction solution (n-heptane:isopropanol = 1:1) was added for ultrasonic crushing, the suspension was centrifuged at 8000× *g* and 4 °C for 5 min, and the TG content was determined using a triglyceride content determination kit (Solarbio Science & Technology, Beijing, China).

### 2.5. Transmission Electron Microscopy (TEM)

The control group and different treatments of the chlamydospore-fermented liquid were sampled at 168 h. The sample was immobilized overnight with 2.5% glutaraldehyde; subjected to agar pre-embedding; cleaned with phosphate buffer solution for 30 min; fixed with osmium for 1.5 h; washed with buffer for 30 min; subjected to 50%, 70%, 80%, 90%, and 100% ethanol gradient dehydration for 15–30 min each; subjected to acetone replacement for 30 min; impregnated for more than 10 h before embedding; and polymerized at 60 °C for 48 h. Uranium staining was performed for 15 min, and lead staining was performed for 5 min after block repair and ultra-thin sectioning. The sections were visualized using TEM.

### 2.6. Confocal Laser-Scanning Microscope

After 168 h of fermentation, samples from each group were obtained, and fat staining was performed according to a previously described method [23]. In short, we prepared a 5% oil red O stock solution, mixed it evenly according to the proportion of oil red O:tri-distilled water (3:2), filtered it using filter paper, and placed it at room temperature for 10 min, showing a wine-red color and no precipitation. Each sample was stained with oil red O working solution for 15 min, washed with sterile water, and observed by CLSM.

### 2.7. Effects of Exogenous Regulators on the Fatty Acid Types of the Chlamydospores of T. harzianum T4

Methyl esterification was conducted according to Zhu et al.’s method [24]. In short, we added 1 mL of sodium hydroxide solution of methanol–water (1:1, *V*:*V*) to the total extracted fat, boiled the water bath for 5 min, added 3 mL of 2 mol/L HCl methanol solution, used an 85 °C water bath for 1 h, added 1 mL of n-hexane, shook the solution well and allowed it to stand to form layers, and transferred the organic phase into a new centrifuge tube, followed by blow-drying with nitrogen to obtain the lipid after methyl esterification. We used an Agilent 6890N gas chromatograph equipped with a 5975 series mass spectrometer. The gas chromatograph–mass spectrometer (GC-MS) was equipped with an HP-5 chromatographic column (30 m × 0.25 mm, 0.25 μm film thickness) for analysis. We set the temperature program to 140 °C for 3 min, increased the temperature to 210 °C at a rate of 3 °C/min for 5 min, and then increased the temperature to 280 °C at 40 °C/min for 3 min. Helium was used as a carrier gas, and the injection volume was 1 μL. We measured the intracellular fatty acid content in chlamydospores after different treatments and determined the proportion of unsaturated fatty acids with an unsaturated index (the unsaturated index is the ratio of unsaturated fatty acids to saturated fatty acids). Data were obtained from three independent repeated tests, where the total methyl ester content of four fatty acids was set to 100%, and the results were determined by the peak area of methyl esters for each fatty acid.

### 2.8. qRT-PCR

To further verify the effect of exogenous regulators on the lipid metabolism of chlamydospores, we studied the expression levels of the lipid metabolism genes *TGL2* and *OLE1*. Since chlamydospore production began at 72 h, mycelium (72 h) was used as the sample. The mycelium was filtered using four layers of sterilized gauze; RNA was extracted with Trizol after grinding in liquid nitrogen and immediately reverse-transcribed into cDNA. Each group consisted of three biological repeats; the selected gene primers are shown in Table 1 (the *18SrRNA* gene was used as the internal reference). The qRT-PCR reaction conditions were as follows: 95 °C for 2 min, 40 cycles of 95 °C for 15 s, and 60 °C for 15–30 s. The dissolution curve was automatically set according to the instrument, and the 2^−^^ΔΔCt^ method was used to analyze the data. The qRT-PCR instrument was a Bio-Rad CFX96 (Bio-Rad, Hercules, CA, USA), and the reverse transcription and fluorescence kits were purchased from Takara (Takara Biotechnology Co., Beijing, China).

### 2.9. Statistical Analysis

All data are from three independent repeated experiments. One-way analysis of variance (ANOVA) followed by Duncan tests was conducted using SPSS Statistics software (V.21.0, SPSS Inc., Chicago, IL, USA). All results were considered statistically significant at *p* < 0.05.

## 3. Results

### 3.1. Induction of Chlamydospores by Exogenous Regulators

We tracked the production of chlamydospores during the fermentation process. Chlamydospores were produced within 48–72 h (3.83 × 10^6^ cfu/mL) and then entered the exponential growth stage; the chlamydospore production peaked at 168 h (2.25 × 10^7^ cfu/mL). Therefore, we measured the yield of chlamydospores at 168 h to study the effects of exogenous regulators on the production of chlamydospores. We found that the addition of glycine betaine, mannitol, and N-A-G, with a certain concentration, has different positive regulatory effects on chlamydospore production of *T. harzianum* T4. The yield of chlamydospores was 5.03 × 10^7^ cfu/mL and 4.67 × 10^7^ cfu/mL after the addition of 2% mannitol and 1% N-A-G, which was 2.27 and 2.10 times higher, respectively, than that of the control group (2.22 × 10^7^ cfu/mL). After the addition of 2% glycine betaine, the yield of chlamydospores increased significantly, reaching 5.05 × 10^7^ cfu/mL, which was 2.28 times higher than that of the control group. The addition of different trehalose concentrations did not affect the chlamydospore yield (Figure 1).

### 3.2. The Stress Resistance of Chlamydospores

We verified the tolerance of chlamydospores induced by different exogenous regulators to UV, heat, and hypertonic stresses. After heat treatment for 7 min, the survival rate of the conidia was only 13.50%, whereas the survival rate of the chlamydospores in all groups was higher than that in the conidia group. Furthermore, compared to the ordinary chlamydospores produced in the control group, the survival rates of chlamydospores in the mannitol and trehalose treatment groups were 51.57% and 40.92%, respectively, which were significantly higher than those in the other groups (Figure 2A).

After UV irradiation for 80 s, the survival rate of the chlamydospores in response to mannitol addition was 46.17%, which was higher than 25.15% in the control group. Surprisingly, the survival rate of the chlamydospores produced by the addition of trehalose was as high as 82.05%. Glycine betaine treatment and N-A-G treatment had no significant effect on the UV resistance of the chlamydospores (Figure 2B).

The chlamydospores produced by the addition of mannitol, glycine betaine, and trehalose had stronger tolerance to hypertonic stress compared to the control group. Under 6% NaCl stress, the germination rate of chlamydospores in the control group was 35.58%, while that in the mannitol, glycine betaine, and trehalose groups was 48.71%, 49.09%, and 81.30%, respectively. Furthermore, the germination rate of the trehalose group reached 7.90% under 9% NaCl stress (Figure 2C). In conclusion, the addition of mannitol and trehalose significantly improved the ability of chlamydospores to resist adverse environmental conditions.

### 3.3. Effects of Exogenous Regulators on Lipid Accumulation in Chlamydospores of T. harzianum T4

We observed the morphological changes in chlamydospores through TEM. First, the TEM images showed that the chlamydospores produced in each group contained a typical double-layer wall structure. The outer wall was thin, and there was a diffuse outer layer outside the outer wall. The inner wall was thick, located between the outer wall and the cell plasma membrane, and the cells contained accumulated fat droplets. Additionally, the accumulation of lipid droplets in chlamydospores under mannitol and trehalose treatments was higher than that in the control group (Figure 3). CLSM pictures confirmed this result (Figure 4). The intracellular lipids were observed as red spots under CLSM after oil red O staining. We could see that the lipid content in the chlamydospores produced by fermentation with mannitol and trehalose was significantly higher.

Therefore, we measured the lipid content in chlamydospores and found that mannitol and trehalose significantly increased lipid accumulation compared to the control group. The total lipid content increased by 36.27% and 14.23% after the addition of mannitol and trehalose, respectively, compared to the control group. Glycine betaine and N-A-G had no significant effect on the lipid content (Figure 5A). We found that the TG content of the control group was 4.25 × 10^−5^ mg/10^4^ cells, whereas the TG content after mannitol and trehalose treatment was 5.90 × 10^−4^ mg/10^4^ cells and 3.63 × 10^−4^ mg/10^4^ cells, respectively; it was 13.88 times and 8.54 times higher than that in the control group (Figure 5B).

Considering these results, we verified by qRT-PCR that the expression of *TGL2*, a gene regulating lipid metabolism, is significantly upregulated after the addition of mannitol and trehalose (Figure 6A). Furthermore, the addition of mannitol and trehalose significantly increased *OLE1* gene expression (Figure 6B). *OLE1* genes improve the tolerance of microorganisms to environmental stress by improving the degree of unsaturated fatty acid [25,26]. These results are consistent with the conclusion that mannitol and trehalose significantly improve the survival rate of chlamydospores under various environmental stresses.

### 3.4. Effects of Exogenous Regulators on Fatty Acid Types of Chlamydospores of T. harzianum T4

To further evaluate whether the exogenous regulators impact the types of fatty acids accumulated by chlamydospores, we measured the fatty acid content in cells using a gas chromatograph–mass spectrometer (GC-MS). We found that the unsaturated index of the mannitol and trehalose groups increased to 2.809% and 2.502%, respectively (Table 2). Interestingly, the proportional changes in each fatty acid type in the chlamydospores were similar after the addition of mannitol and trehalose; the palmitic acid content decreased significantly to 19.577% and 21.498%, respectively, and the linoleic acid content increased significantly to 35.419% and 34.771%, respectively. Our results showed that these two regulators are involved in the lipid-related metabolic pathway during chlamydospore production and significantly increase the degree of unsaturated intracellular fatty acids.

## 4. Discussion

We found that the addition of different exogenous regulators during liquid fermentation has different regulatory effects on the formation of chlamydospores in *T. harzianum* T4. Mannitol, glycine betaine, and N-A-G effectively increased the yield of chlamydospores. Although they are often used to help microorganisms resist adverse environments [27,28,29,30,31,32], they also participate in the spore production process. Chitin is an important component of the cell wall of chlamydospores, and N-A-G is a monomer of chitin. It has been found that N-A-G specifically induces the production of chlamydospores in *Candida albicans* [33]. It also affects the transformation between mycelia and yeast-like cells [34,35,36]. *Gibberella zeae* produces chlamydospore-like structures, and the mannitol level increases during the formation process [17]. Glycine betaine is mainly used as an organic osmotic protector or methyl donor. In the face of various environmental stresses, cells accumulate glycine betaine [37,38], and more glycine betaine is used in lipid biosynthesis to maintain membrane fluidity [19,39]. These studies also indicate that mannitol, glycine betaine, and N-A-G are important raw materials for assembling and producing chlamydospores. Therefore, we suggest that they enhance chlamydospore production in *T. harzianum* T4 liquid fermentation by providing important chlamydospore components.

In addition to regulating chlamydospore production, we found that chlamydospores produced by the addition of mannitol and trehalose during liquid fermentation have stronger resistance to heat, UV, and hypertonic stresses. Furthermore, the addition of mannitol and trehalose increased the intracellular lipid content and degree of unsaturated fatty acids in chlamydospores. Unsaturated fatty acids maintain the integrity and function of the cell membrane and adapt to environmental stress [40]. Studies have shown that the mannitol synthesis and fatty acid pathways compete for carbon storage in *Yarrowia lipolytica* [41]. In the yeast *Rhodosporidiobolus fluvialis*, mannitol was reported to modulate fatty acid synthesis and increase polyunsaturated fatty acids compared to glucose [42]. These findings suggest that the presence of mannitol is important for regulating the lipid metabolism of chlamydospores. Trehalose is often used as an osmotic pressure regulator [43,44] and a protective drying agent [45] and maintains the stability of liposomes under pressure [46]. Trehalose can also be used as a unique carbon source for fungal growth and lipid synthesis, acting as a protective agent of fatty acids [47]. In a study on *Aspergillus niger*, the response to hypertonic stress and heat shock included the accumulation of trehalose and mannitol, with an increased phosphatidic acid proportion in membrane lipids [48].

TGL2 is a triacylglycerol lipase with certain lipolysis activity toward triacylglycerol. However, in a study on *Nannochloropsis* by Nobusawa et al., *ΔTGL2* accumulated triacylglycerol under nutrient depletion and the triacylglycerol was degraded rapidly after transfer to a fresh medium; thus, TGL2 is not likely to be engaged in bulky degradation of triacylglycerol [49]. JinHam et al. proposed that *TGL2* encodes a functional lipase to increase the concentration of oleic acid in *Saccharomyces cerevisiae* mitochondria and is involved in lipolysis catalyzed by the mitochondrial TGL2 lipase, and its likely product oleic acid may become crucial for the survival of cells under stress conditions [50]. The *OLE1* gene regulates the production of fatty acid desaturase in microorganisms and acts on biological membranes to maintain their fluidity and permeability [40,51]. The *OLE1* gene also regulates the content and composition of unsaturated fatty acids in *S. cerevisiae*, which helps cells resist lipid peroxidation caused by high temperatures [24,52]. In our study, chlamydospores were produced in large quantities 72 h onward and the expression of chlamydospore-related genes was more intense, so we selected mycelium at 72 h for gene expression study. The results showed that although *TGL2* expression was significantly upregulated in the early stages of chlamydospore production, a significant accumulation of TG was still detected. Therefore, we determined that mannitol and trehalose upregulate *TGL2* expression significantly, which may be more involved in fatty acid-type changes. The expression changes of *OLE1* as a fatty acid desaturase confirmed this finding. This evidence suggests that trehalose and mannitol enhance the stress resistance of chlamydospores by promoting lipid accumulation and increasing the degree of unsaturated fatty acids. Peng et al. conducted transcriptome analysis at different *Trichoderma virens* chlamydospore production stages. They found that the fatty acids were degraded to provide nutrients for growth in the early sporulation stage; in the late sporulation stage, fatty acid biosynthesis genes were upregulated to promote lipid accumulation in chlamydospores [53]. We believe that lipid metabolism is an important part of chlamydospore production and lipid metabolism adjustment helps chlamydospores cope with more complex environments, which impacts the stress resistance of chlamydospores.

In conclusion, mannitol, glycine betaine, N-A-G, and trehalose all affect the production of chlamydospores and their states differently. Mannitol, glycine betaine, and N-A-G significantly increase the yield of chlamydospores. At the same time, mannitol and trehalose improve the tolerance of chlamydospores to various stresses by changing lipid metabolism. Our research also showed that although chlamydospore formation is a self-protection strategy adopted against external adverse growth environmental stress, it can still be induced by the participation of external regulators under normal culture conditions. We believe our research will provide new ideas for exploring spore production in filamentous fungi and will guide the shelf-life improvement of preparations containing biocontrol fungi.

## Figures and Tables

**Figure 1 jof-08-01017-f001:**
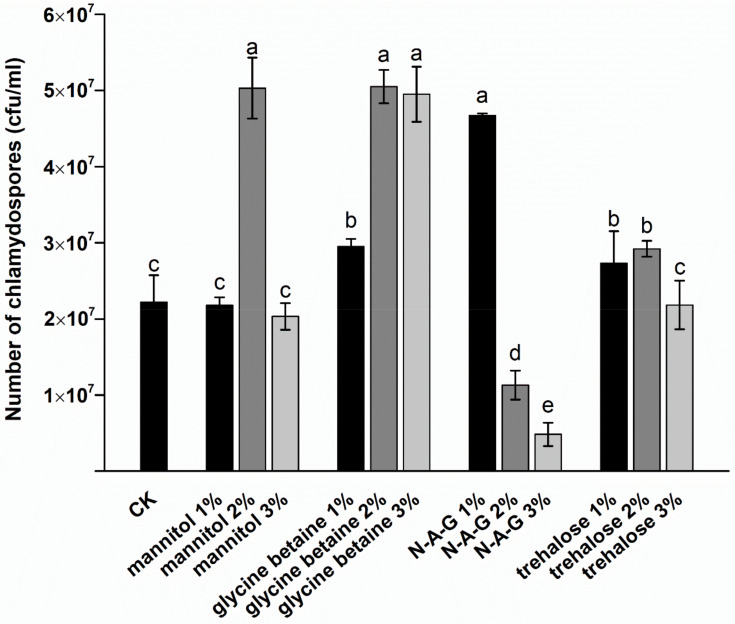
Effects of different exogenous regulators on the chlamydospore yield. N-A-G: N-acetylglucosamine. The data are from three independent repeated experiments. The different letters above the bars indicate a significant difference in the chlamydospore yield for each treatment, according to the least significant difference test (*p* < 0.05).

**Figure 2 jof-08-01017-f002:**
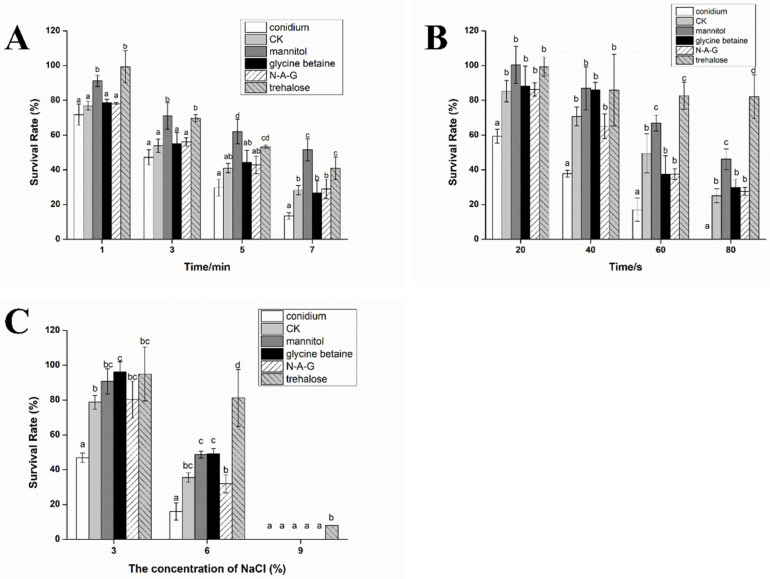
Stress resistance ability of chlamydospores produced by adding different exogenous regulators: (**A**) heat stress, (**B**) UV stress, and (**C**) hypertonic stress. The data are from three independent repeated experiments. The different letters above the bars indicate that the survival rates of chlamydospores produced by adding different exogenous regulators are significantly different when facing different environmental pressures, according to the least significant difference test (*p* < 0.05).

**Figure 3 jof-08-01017-f003:**
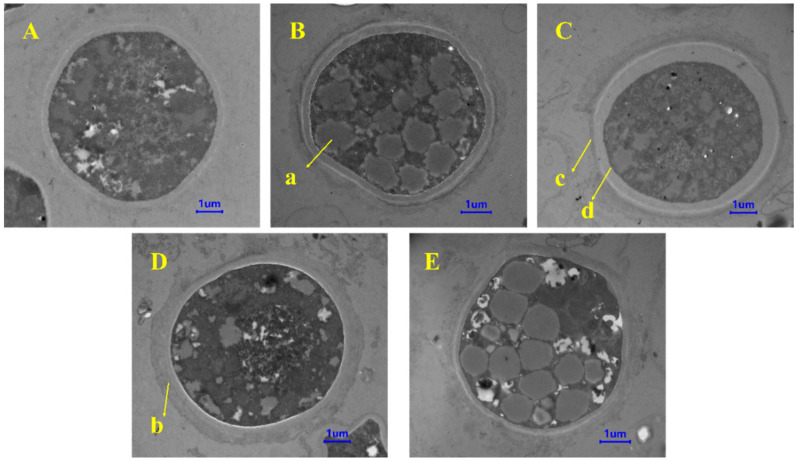
Transmission electron microscopy (TEM) images of chlamydospores: (**A**) control, (**B**) mannitol, (**C**) glycine betaine, (**D**) N-A-G, and (**E**) trehalose. The scale bar is 1 μm. (**a**) Fat droplet, (**b**) diffuse outer layer, (**c**) outer wall, and (**d**) inner wall.

**Figure 4 jof-08-01017-f004:**
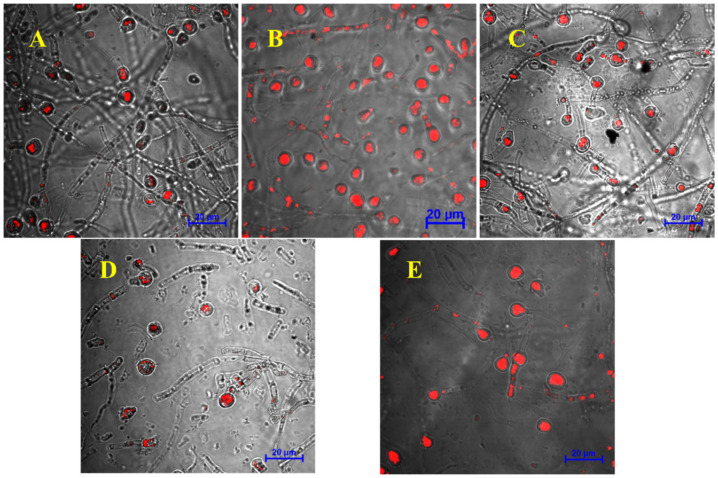
Images of chlamydospores produced by adding different exogenous regulators, as observed under a confocal microscope: (**A**) control group, (**B**) mannitol, (**C**) glycine betaine, (**D**) N-A-G, and (**E**) trehalose. The scale of the image is 20 μm.

**Figure 5 jof-08-01017-f005:**
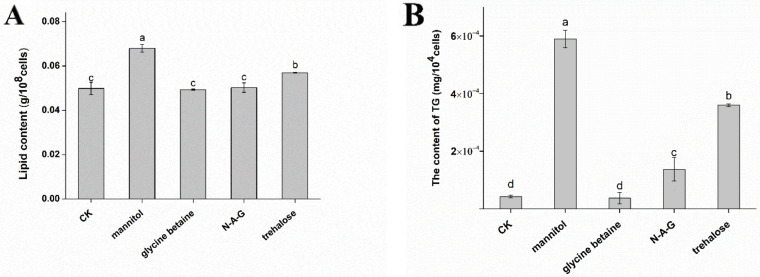
Effects of different exogenous regulators on the total lipid and triglyceride accumulation: (**A**) lipid content and (**B**) TG content. The data are from three independent repeated experiments. The different letters above the bars indicate a significant difference in the lipid content or TG content for each treatment, according to the least significant difference test (*p* < 0.05).

**Figure 6 jof-08-01017-f006:**
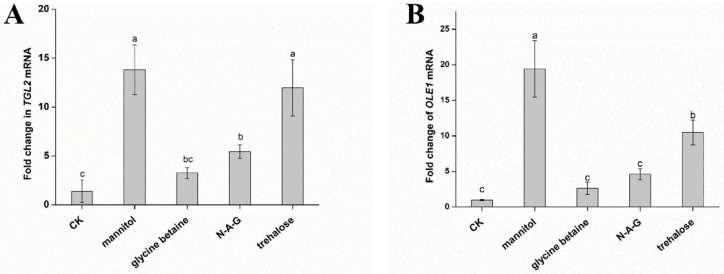
Differences in the expression of genes related to lipid metabolism caused by different exogenous regulators: (**A**) *TGL2* gene and (**B**) *OLE1* gene. The data are from three independent repeated experiments. The different letters above the bars indicate a significant difference in the expression level of the *TGL2* gene and the *OLE1* gene for each treatment, according to the least significant difference test (*p* < 0.05).

**Table 1 jof-08-01017-t001:** Primer sets used in PCR.

Primer Name	Sequence (5′–3′)
*TGL2*	CCGCCGTCAACATCATCG
	ACCAGGTAATCCGCAAAGC
*OLE1*	ATGTCCGAACCCACAGCCTC
	TTAAGCAGCGTCGGCGCTG
*18SrRNA*	GCAACGAGTAAAGCACCAGA
	GGGGTTCATTCGTGTGTAGC

**Table 2 jof-08-01017-t002:** Analysis of the fatty acid composition and content in chlamydospores in response to different regulators. The values indicate the mean ± standard error (SE). The data are from three independent experiments. The different letters after the values in each row represent significant differences in the content of fatty acids in the chlamydospores produced by adding different exogenous regulators.

	Fatty Acid Type	Number of Carbon Atoms	Control (%)	Mannitol (%)	Glycine Betaine (%)	N-A-G (%)	Trehalose (%)
1	Palmitic acid	C16:0	39.049 ± 1.296 a	19.577 ± 0.634 c	33.497 ± 0.766 b	33.039 ± 0.819 b	21.498 ± 1.885 c
2	Linoleic acid	C18:2	9.171 ± 0.293 c	35.419 ± 0.857 a	14.187 ± 1.138 b	10.037 ± 0.969 c	34.771 ± 0.117 a
3	Oleic acid	C18:1	32.246 ± 1.949 c	38.309 ± 1.165 ab	36.717 ± 1.504 b	40.081 ± 1.513 a	36.561 ± 2.183 b
4	Stearic acid	C18:0	19.533 ± 0.959 a	6.695 ± 0.25 c	15.599 ± 0.198 b	16.81 ± 0.252 b	7.17 ± 0.397 c
5	Unsaturated index		0.709 ± 0.066 d	2.809 ± 0.124 a	1.039 ± 0.084 c	1.006 ± 0.03672 c	2.502 ± 0.266 b

## Data Availability

Not applicable.

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
