# Peer review of "Exogenous Regulators Enhance the Yield and Stress Resistance of Chlamydospores of the Biocontrol Agent Trichoderma harzianum T4"

_jof, 2022, doi:10.3390/jof8101017_

Round 1

Reviewer 1 Report

General comments

This work is  important from the practical point of view.  It can improve Trichoderma  biocontrol potential.

Using various methods,  the  Authors  proved that mannitol and trehalose stimulate the production of chlamydospores and enhance their resistance to stress factors. The obtained results show that the addition of exogenous regulators affects lipid  metabolism  of chlamydospores and  enables  better adaptation to environmental stress.

Detailed comments

In the Materials and Methods section, please add where you purchased the chemical reagents used in the work.

Please complete what statistical tests were carried out in the work.

The Authors should   provide more information about the kinetics of chlamydospore production. In the genetic part of the work,
72 h mycelium was used for the study
of gene expression: "chlamydospore production began at 72 h" (line 158).
In the other parts of the study chlamydospores from 168 h of culture were examined.
Please explain this difference.

Author Response

We are grateful to you for spending your precious time and energy reviewing our manuscript and providing us with valuable suggestions. 

Below are our specific point-by-point responses to the comments and suggestions.

1. “In the Materials and Methods section, please add where you purchased the chemical reagents used in the work.”

We thank you for your suggestion. We have supplemented the sources of chemical reagents in the Materials and Methods section.

2. “Please complete what statistical tests were carried out in the work.”

We thank you for your suggestion. We have added a statistical analysis section to the article.

3. “The Authors should provide more information about the kinetics of chlamydospore production. In the genetic part of the work, 72 h mycelium was used for the study of gene expression: "chlamydospore production began at 72 h" (line 158). In the other parts of the study chlamydospores from 168 h of culture were examined. Please explain this difference.”

We appreciate your suggestion.  We have tested the kinetics of chlamydospore production in our previous work, due to the space without detailed explanation in the manuscript. The results showed that chlamydospores were produced in large quantities from 72h (3.83 × 106 cfu/mL), the expression of chlamydospore related genes was more intense, so we selected mycelium at 72h for gene expression study. Chlamydospore mature and yield peaks at 168h (2.25 × 107 cfu/mL), so chlamydospores at 168h were selected for blood cell count and stress resistance testing. We have made a brief explanation in the Result section (Line 179) and Discussion section (Line 329) of the article.

Reviewer 2 Report

This manuscript reports on effects of some regulators on chlamydospore production by a strain of Trichoderma harzianum, introducing applicative perspectives in the production of preparations containing biocontrol fungi with increased shelf life. It is concise and properly organized, and in my opinion only requires a general improvement of the language style. In this respect, below I provide a list of minimum corrections needed to make it acceptable.

Line 11: delete 'Trichoderma'; line 14: delete '(N-A-G)', as this abbreviation is only to be indicated in the main text; line 16: delete '(UV)', same reason as above; line 26: correct to 'chlamydospores in filamentous fungi'; line 37: delete ', Fusarium oxysporum';
lines 42-43: delete 'Mycelia or conidia transform into chlamydospores, a stress-resistant strategy, after external and internal stresses', as this concept is expressed at the next line;
line 44: correct 'a slowing' to 'slow'; lines 47-49: make this sentence more essential, such as 'Therefore, a preparation that comprises chlamydospores may guarantee a prolonged shelf life.' line 50: as this is the first mention in the main text, use again 'N-acetylglucosamine' and introduce its abbreviation 'N-A-G' here; lines 54-55: change to '..., while N-A-G represents the fundamental product for chitin biosynthesis.'; line 58: change to '...; hence, lipid metabolism is fundamental in...'; lines 59-60: delete 'Therefore, we investigated whether these exogenous regulators promote the formation process and affect the resilience of chlamydospores.' since this concept is repeated at the next paragraph;
lines 67-68: change to '...the differentiation pathway of chlamydospores in filamentous fungi and a basis for obtaining industrial preparations with increased chlamydospore content.' M&M: text in this section is generally correct; however, I recommend a revision in view of making it more fluent. As an example, text at lines 91-93 could be adjusted as '...: the fermentation liquid was vortexed for 5 min, followed by ultrasonic treatment for 10 min; mycelium was removed by filtration through double-layer lens-wiping paper.'; line 192: start new line after '(Figure 2A)'; line 197: start new line after '(Figure 2B)';   line 204: correct 'environments' to 'environmental conditions'; line 219: authors should be more circumstantial in describing how CLSM images confirm results of TEM observations; lines 242-244: change to '...that expression of TGL2, a gene regulating lipid metabolism, was significantly upregulated after the addition of mannitol and trehalose (Figure 6A).'; line 246: correct to 'microorganisms' and 'stress'; lines 296-298: these sentences should be separated: correct to '...lipolytica [42]. In the yeast Rhodosporidiobolus fluvialis mannitol was reported to modulate fatty acid synthesis and to increase polyunsaturated fatty acids as compared with glucose [43].
line 304: correct to '... mannitol, with increased phosphatidic...'; line 316: correct to 'S. cerevisiae';
line 324: delete 'Xinhong'; line 336: add comma after 'that'; line 337: correct 'for' to 'against'; lines 339-340: correct to '... exploring spore production in filamentous fungi and will guide the shelf-life improvement of preparations containing biocontrol fungi.'

Author Response

We are grateful to you for spending their precious time and energy reviewing our manuscript and providing us with valuable suggestions. 

Below are our specific point-by-point responses to the comments and suggestions.

  1. “In my opinion only requires a general improvement of the language style. In this respect, below I provide a list of minimum corrections needed to make it acceptable.”

We thank you for your careful reviewing. We agree with all the  suggestions and have carefully revised our manuscript in revision mode according to your suggestions.

  1. “Line 219: authors should be more circumstantial in describing how CLSM images confirm results of TEM observations”

We appreciate your suggestion. We have further elaborated the mutual confirmation of CLSM and TEM in the Results section (Line227) of the manuscript.